# *Divide et impera*: An *In Silico* Screening Targeting HCMV ppUL44 Processivity Factor Homodimerization Identifies Small Molecules Inhibiting Viral Replication

**DOI:** 10.3390/v13050941

**Published:** 2021-05-20

**Authors:** Hanieh Ghassabian, Federico Falchi, Martina Timmoneri, Beatrice Mercorelli, Arianna Loregian, Giorgio Palù, Gualtiero Alvisi

**Affiliations:** 1Department of Molecular Medicine, University of Padova, 35121 Padova, Italy; haniehghassabian@gmail.com (H.G.); martina.timmoneri@gmail.com (M.T.); beatrice.mercorelli@unipd.it (B.M.); arianna.loregian@unipd.it (A.L.); giorgio.palu@unipd.it (G.P.); 2Molecular Horizon, 06084 Bettona, Italy; federico.falchi@hotmail.com

**Keywords:** HCMV, protein-protein interactions, small molecules, ppUL44, PAP, pUL54, antivirals, screening

## Abstract

Human cytomegalovirus (HCMV) is a leading cause of severe diseases in immunocompromised individuals, including AIDS patients and transplant recipients, and in congenitally infected newborns. The utility of available drugs is limited by poor bioavailability, toxicity, and emergence of resistant strains. Therefore, it is crucial to identify new targets for therapeutic intervention. Among the latter, viral protein–protein interactions are becoming increasingly attractive. Since dimerization of HCMV DNA polymerase processivity factor ppUL44 plays an essential role in the viral life cycle, being required for oriLyt-dependent DNA replication, it can be considered a potential therapeutic target. We therefore performed an *in silico* screening and selected 18 small molecules (SMs) potentially interfering with ppUL44 homodimerization. Antiviral assays using recombinant HCMV TB4-UL83-YFP in the presence of the selected SMs led to the identification of four active compounds. The most active one, B3, also efficiently inhibited HCMV AD169 strain in plaque reduction assays and impaired replication of an AD169-GFP reporter virus and its ganciclovir-resistant counterpart to a similar extent. As assessed by Western blotting experiments, B3 specifically reduced viral gene expression starting from 48 h post infection, consistent with the inhibition of viral DNA synthesis measured by qPCR starting from 72 h post infection. Therefore, our data suggest that inhibition of ppUL44 dimerization could represent a new class of HCMV inhibitors, complementary to those targeting the DNA polymerase catalytic subunit or the viral terminase complex.

## 1. Introduction

The *β-Herpesvirinae* member *human cytomegalovirus* (HCMV) is a major human pathogen, causing severe and life-threatening infections in immunocompromised subjects [1] and in congenitally infected newborns [2]. Herpesviruses are opportunistic double-stranded DNA viruses, whose genome transcription, replication, and encapsidation occur in the host cell nucleus [3]. The molecular mechanisms involved in herpesvirus DNA replication and its regulation have been widely studied as they provide important models for the study of eukaryotic DNA replication and because viral enzymes involved in the process represent targets for antiviral therapy. HCMV DNA polymerase holoenzyme is a multi-functional enzyme that plays a key role during viral infection ensuring replication of the viral genome, and consists of the catalytic subunit pUL54 and the processivity factor ppUL44, which physically and functionally interact thought specific residues [4,5,6]. Not surprisingly, the most widely antiviral agents used to fight HCMV infections target pUL54 and are either nucleoside or pyrophosphate analogues such as ganciclovir (GCV) or foscarnet (PAA), respectively [7]. However, long-term administration of these antiviral agents frequently leads to the selection of viral isolates with reduced drug susceptibility, due to mutations of either pUL54 or of pUL97, the viral kinase phosphorylating GCV [8,9]. Treatment with the recently approved Letermovir, which targets the viral terminase complex [10,11], has been similarly shown to cause the selection of viral resistant strains [12,13]. Therefore, there is a recognized need for novel anti-HCMV compounds that target other viral functions [14].

Intriguingly, inhibition of either ppUL44 expression or its interaction with pUL54 strongly impairs HCMV replication, suggesting that it may represent a potential alternative antiviral target [15,16,17]. Indeed, ppUL44 can directly bind to dsDNA and pUL54, thus tethering the DNA polymerase holoenzyme to the DNA template [18,19]. While the N-terminal domain of ppUL44 (residues 1-290) retains all known ppUL44 biochemical properties [20], its C-terminal domain is the target of several post translational modifications which modulate protein nuclear import [21,22]. Despite low sequence homology, ppUL44 N-terminal domain displays a similar fold to PCNA [23]. However, in stark contrast with the trimeric PCNA, ppUL44 (1-290) forms head-to-head dimers, adopting a C-clamp-shaped structure. Dimerization relies on six main-chain-to-main-chain hydrogen bonds and extensive packaging of hydrophobic side chain at the interface, enabling the formation of a central, positively charged cavity which binds the viral DNA via electrostatic interactions [23,24]. Accordingly, ppUL44 dimerization is important for viral DNA binding: substitution of residues at the dimerization interface such as L86 and L87, which make extensive contacts with the hydrophobic residues along the dimer interface, strongly impairs both ppUL44 dimerization and dsDNA binding in vitro [23]. Furthermore, the ppUL44-dsDNA interaction also depends on basic residues located within a highly flexible “gap loop” not visible in the published crystal structure, which are involved in additional electrostatic interactions with the dsDNA backbone [24,25]. Intriguingly, recent data from our laboratory suggest that dsDNA binding of ppUL44 is essential for HCMV DNA replication, since substitutions either within the basic gap loop or at the dimerization interface abolished the ability of ppUL44 to trans-complement oriLyt dependent DNA replication, without affecting other ppUL44 biochemical properties [25,26,27,28]. Therefore, ppUL44 dimerization is required for tethering the DNA polymerase holoenzyme to viral dsDNA and thus represents an attractive antiviral target [28].

Protein–protein interactions (PPIs) are essential to all biological processes and can be modulated by small molecules [29,30,31], thus representing a large class of therapeutic targets and implying the possibility to impair viral replication and pathogenesis [32,33,34]. Several inhibitors have reached the clinical trials thanks also to the development of computational and chemical technologies, alongside with experimental and virtual fragment screens used to define the druggability of PPIs [35,36]. Most of such inhibitors target PPIs in which partner proteins are characterized by short primary sequences at the interface [37,38] and the hot-spots residues are concentrated in small binding pockets [39,40]. Recent studies have also reported the inhibition of protein dimerization, with a focus on proteins overexpressed in cancer or involved in viral replication [30,41,42,43].

In this study, we aimed at identifying small molecules (SMs) that can hinder HCMV replication by interfering with ppUL44 homodimerization. To this end, we used the ppUL44 homodimer structure as a template to screen *in silico* about 1.3 million SMs from the ZINC database and identify 18 SMs potentially able to interfere with ppUL44 homodimerization. Our results clearly showed that one out of the 18 SMs tested (i.e., SM B3) could inhibit replication of different HCMV strains, including a GCV-resistant strain, as assessed by plaque, fluorescence, and virus yield reduction assays. Finally, B3 strongly inhibited HCMV DNA replication with a kinetic similar to GCV, and affected viral gene expression only starting from 48 h post infection (p.i.), causing a strong decrease of late gene pp28 expression at 72 h and 96 h p.i., consistent with interference with ppUL44 homodimerization.

## 2. Materials and Methods

### 2.1. Analysis of ppUL44 Dimerization Interface

The crystallographic structure of ppUL44 homodimer [23] was downloaded from the Protein Data Bank (pdb code: 1T6L). Only the A chain was extracted from the complex. The structure was optimized with the Protein Preparation Wizard tool of the Schrödinger suite (Schrödinger). The presence of potentially druggable sites at the dimerization interface of ppUL44 (1-290) was assessed with software Sitemap [44], using default settings apart from grid resolution which was set as fine.

### 2.2. Virtual Screening Database Preparation

A 3D molecular database was built with the Schrödinger suite (Schrödinger) starting from 2D structures taken from the ZINC database (www.zinc.docking.org; accessed on 1 March 2013). A total number of about 3.6 million compounds were selected (Vendors: Asinex, Chembridge, Princeton, NCI and ZINC natural) and downloaded for this study. The 2D structures were converted into 3D structures and stereoisomers were generated with the Ligprep function of the Schrödinger suite (Schrödinger). Moreover, for each entry all the possible ionization states at pH 7.0 ± 2.0 and tautomers were generated with Epik (Schrödinger). The obtained database consisted of about 5 million compounds. In order to retrieve the most drug-like compounds, ADMET properties of each molecule in the database were predicted with Qikprop (Schrödinger) and compounds were filtered as described in [45] using a “soft Lipinsky rule” (Molecular weight ≤ 600, Rotable bonds ≤ 10, Number of H-bond acceptors ≤ 10, Number of H-bond donors ≤ 5, Number of chiral centers ≤ 2, QplogPo/w ≤ 6). Finally, the number of compounds was reduced to about 1.3 million by using the PPI-HitProfiler software with the “soft” mode (CDithem; http://www.cdithem.fr; accessed on 15 March 2013).

### 2.3. Virtual Screening

For the docking stage, a receptor grid was built on the ppUL44 structure prepared as above. The grid was centered on the position of residue M116 and receptor grid generation default settings were applied. By using this grid, the database was docked with the High throughput virtual screening (HTVS) scoring function of the Glide software (Schrödinger). After this run, we selected 50,000 compounds on the basis of the HTVS docking score. Selected molecules were submitted to a second run of docking using the standard precision (SP) scoring function and only the 5000 top-ranked compounds were selected. Finally, selected molecules were submitted to a run of docking using the extra precision (XP) scoring function and only the 500 top-ranked compounds were selected. After docking, we further narrowed down the number of selected compounds to 18 on the basis of commercial availability, visual inspection, and by a cluster analysis performed with Tanimoto on the basis of the Molprint2D fingerprints of each molecule [46].

### 2.4. Cell Lines

Human foreskin fibroblast (HFF), MRC5 (#CCL171, ATCC), and HEK 293T (#3216. ATCC) cells were maintained in Dulbecco’s modified Eagle’s medium (DMEM) supplemented with 10% (*v*/*v*) foetal bovine serum (FBS), 50 U/mL penicillin, 50 U/mL streptomycin, and 2 mL-glutamine and passaged when reached confluence. MCR5 cells were used up to passage number 30 and subsequently discarded.

### 2.5. Viruses, Viral Stocks Preparation and Titration

HCMV laboratory strain AD169 was obtained from ATCC (ATCC, #VR-53). Recombinant virus AD169-GFP, expressing a humanized version of GFP under control of the HCMV IE promoter between open reading frames US9 and US10, as well as its GCV-resistant derivative AD169-GFP26, bearing the UL97 M460I substitution [47], were a generous gift by Manfred Marschall (Erlangen, Germany). Recombinant virus TB4-UL83-YFP, wherein YFP is fused to the C-terminus of the early-late tegument phosphoprotein pp65 [48] was kindly provided by Michael Winkler (Gottingen, Germany). All viral stocks were prepared and titered by immunological detection of IE1/2 proteins as described in [47,48] or by the plaque reduction method [49]. For viral stocks preparation, MRC5 cells were seeded in T150 flasks (6 × 10^6^ cells/flask) the day before infection, which was carried in the absence of serum, at a multiplicity of infection (MOI) of 0.02–0.05 infectious units (IU)/cell, for 2 h in a humidified incubator at 37 and 5% CO_2_ with occasional shaking. Cell medium was changed at the end of the incubation time and every 3 days. Viral supernatants were collected 7 days p.i., centrifuged for 5′ at 700 rpm to remove cell debris, and stored in the −80 °C after addition of 1% (*v*/*v*) DMSO. For titration of infectivity by immunological detection of IE proteins, MRC5 cells were seeded in 96-well plates (1.5 × 10^4^ cells/well) in 200 mL of DMEM/well. The day after, viral stocks were rapidly thawed at 37 °C as described above. Serial 10-fold dilutions of viral stocks were used to infect three wells per dilution for 2 h at 37 °C in a final volume of 200 μL. Twenty-four h later, the media was removed and the cells washed once with PBS (200 μL/well). Subsequently cells were fixed for 15 min with EtOH 95% at RT followed by saturation of unspecific binding sites by incubation with PBS containing 5% FBS (*w*/*v*) for 1 h at 37 °C (50 μL/well). After 3 washes with PBS, cells were incubated with α-IE1&2 mAb (#CA003-1, Virusys Corporation, Taneytown, MD, USA; 1:100) diluted in PBS containing 5% FBS for 16 h at +4 °C. Cells were washed 3× with PBS and incubated with either Alexa Fluor 488 Alexa Fluor 555 the goat anti-mouse IgG secondary antibodies (#A-11001 and #A-21424, respectively, Themofisher Scientific, Waltham, MA, USA; 1:1000) diluted in PBS containing 5% FBS for 1 h at RT. After 3× washes with PBS positive cells were visualized and counted using an inverted fluorescent microscope (DMIL, Leica, Leica Microsystems, Wetzlar, Germany) equipped with a digital camera (DFC420C, Leica), to allow calculation of the viral titer expressed as IU/mL.

### 2.6. Preparation of Small Molecules (SMs) and Ganciclovir (GCV) Stocks

Ganciclovir (GCV; S1878, Selleckchem, Houston, TX, USA) and small molecules (SMs), with a >90% purity assessed by liquid chromatography mass spectrometry and high-performance liquid chromatography (Vitas-M Laboratory, Radio City, Hong Kong), were resuspended in 100% DMSO to obtain 50 mM and 20 mM stocks, respectively, and stored at −20 °C protected from light.

### 2.7. Antiviral Compounds Testing

For identification of small molecules (SMs) active on HCMV replication, MRC5 cells (1.5 × 10^4^ cells/well) were seeded in 96-well special optics black microplates (CLS3614, Corning, Corning, NY, USA) and incubated overnight at 37 °C, 5% CO_2_, and 95% humidity. The next day, cells were infected for 1 h at 37 °C in 100 μL/well of DMEM containing TB4-UL83-EYFP at an MOI of 0.03 IU/cell. Cells were subsequently washed, and media containing either 0.5% DMSO or two different concentrations (100 and 10 μM) of SMs with a 0.5% DMSO final concentration was added. GCV (50 μM) was included as a positive control for inhibition of viral replication. Mock infected cells served as a reference for calculation of background fluorescence. After addition of SMs, the plates were further incubated at 37 °C, 5% CO_2_ and 95% humidity. Every day, cell confluence and morphology, as well as CPE and the presence of precipitates were evaluated by light microscopy. Fluorescence signals were visualized on an inverted fluorescent microscope (DMIL, Leica). Seven days p.i., cells were washed once with PBS and lysed with luciferase lysis buffer (25 mM glycylglycine, 15 mM MgSO_4_, 4 mM EGTA, 0.1 % Triton X-100, pH 7.8). Plates were immediately frozen at −20 °C, thawed at RT, and fluorescent signals relative to each condition were acquired using a reader compatible with fluorescence measurements (VICTOR X2 Multilabel Plate Reader, Perkin Elmer, Waltham, MA, USA) equipped with a fluorimetric excitation filter (band pass 485 ± 14 nm) and a fluorimetric emission filter (band pass 535 ± 25 nm) as described in [50]. After background subtraction, data were normalized to solvent-treated controls and analyzed with Graphpad Prism (Graphpad Software Inc., San Diego, CA, USA). The screening was performed three times, and each plate included at least two wells treated with the same compound, as well as at least 12 wells treated with solvent only.

### 2.8. Fluorescence Reduction Assays (FRA)

To calculate the effective dose 50 (ED_50_) of each compound against TB4-UL83-EYFP virus by means of FRA, MRC5 cells were seeded, infected, treated, and processed as above, using increasing concentrations of each SMs and GCV (range between 0.02 and 100 μM). Mock infected cells served as a reference for calculation of background fluorescence. After background subtraction, data were normalized to solvent-treated controls and analyzed with Graphpad Prism (Graphpad Software Inc.). The experiments were performed six times, and each plate comprised at least two wells treated with the same compound, as well as at least 14 wells treated with solvent only. To calculate the ED_50_ of each compound against AD169-GFP virus and its GCV-resistant AD169-GFP26 counterpart by the means of FRA, MRC5 cells were seeded in 12-well plates (1.8 × 10^5^ cells/well) in 1 mL DMEM supplemented with 10% (*v*/*v*) FBS, 50 U/mL penicillin, 50 U/mL streptomycin and 2 mM L-glutamine, and incubated overnight at 37 °C, 5% CO_2_ and 95% humidity. The next day, cells were infected for 2 h at 37 °C in 1 mL/well of DMEM at MOI of 0.05 IU/cell as described in [47]. Subsequently, cells were washed with 2 mL of PBS, and 1 mL of medium containing either 0.5% DMSO or increasing concentrations (from 0.001 to 100 μM) of each compound with a 0.5% DMSO final concentration was added. Every day, cell confluence and morphology, as well as CPE and the presence of precipitates, were evaluated by light microscopy, whereas fluorescence signals were visualized on an inverted fluorescent microscope (DFC420 C, Leica). Seven days p.i., supernatants were collected, cleared from cells and debris by centrifugation for 5 min at 700 rpm, and stored at −80 °C until used for virus yield reduction assays (VYRAs) as described below. Cells were washed with 2 mL of ice cold PBS and lysed in 200 μL of GFP lysis buffer (25 mM Tris-HCl, pH 7.8, 2 mM DTT, 2 mM trans-1,2-diaminocyclohexane-*N*,*N*,*N*,*N*-tetraacetic acid, 1% Triton X-100, 10% glycerol (*v*/*v*)). Plates were further incubated 10 min at 37 °C in a humidified incubator, before being incubated for 30 min at RT with shaking at 225 rpm. Samples were centrifuged for 5 min at 4 °C at 13,000 rpm and 100 μL of cleared lysates were transferred to black bottomed 96-well plates (#3916, Corning). Fluorescent signals were acquired and analyzed as described above to calculate the *p*-value relative to the Student’s *t*-test between appropriate groups.

### 2.9. Virus Yield Reduction Assays (VYRAs)

To determine the ED_50_ of each compound against AD169-GFP virus and its GCV-resistant AD169-GFP26 counterpart by means of VYRAs, MRC5 cells were seeded in clear flat bottom 96-well tissue culture plates (1.5 × 10^4^ cells/well) with low evaporation lids (#353072, Corning). The next day, the medium was replaced with serial dilutions of supernatants containing AD169-GFP or AD169-GFP26 virus grown in the presence of inhibitory compounds. One week later, virus yield relative to each condition was calculated and expressed as 50% Tissue Culture Infectious Dose (TCID_50_)/mL using the Spearman and Karber algorithm as described in [51]. Data was statistically analyzed using Graphpad Prism (Graphpad Software Inc.), to calculate the *p*-value relative to the Student’s *t*-test between appropriate groups.

### 2.10. Plaque Reduction Assays (PRA)

The effect of B3 and GCV on AD169 replication in HFF cells was investigated by plaque reduction assays (PRA) as previously described [52]. Briefly, HFF cells were seeded in 24-well plates (2 × 10^5^ cells/well). The following day cells were infected at 37 °C with 70 Plaque Forming Unit (PFU) of HCMV AD169 per well in DMEM containing FBS 5%. At 2 h p.i., the inocula were removed, cells were washed, and media containing various concentrations of each compound, 5% FBS, and 0.6% methylcellulose were added. All compound concentrations were tested at least in triplicate. After incubation at 37 °C for 10–11 days, cell monolayers were fixed, stained with crystal violet, and viral plaques were counted.

### 2.11. Cell Cytotoxicity Assays

To evaluate the effect of SMs on cell viability and proliferative potential, three separate assays were performed. MTT and MTS assays, which measure NAD(P)H-dependent oxidoreductase enzymes and CellTiter-Glo^®^ Luminescent Cell Viability Assay assays, which measure intracellular ATP. For MTT assays, MRC5 cells (1.5 × 10^4^ cells/well) were seeded in clear flat bottom 96-well tissue culture plates with low evaporation lids (#353072, Corning) in duplicate. After 24 h, cells were treated with different concentrations of GCV or SMs, or solvent only. A number of wells containing only DMEM and no cells were also included for background correction. Seven days post-treatment, cells were processed for measurement of cell metabolic activity using 3-(4,5-Dimethyl-2-thiazolyl)-2,5-diphenyl-2H-tetrazoliumbromid (MTT; #A2231, Applichem, Darmstadt, Germany) following the manufacturer’s recommendations. After background subtraction, data were normalized to solvent-treated controls and analyzed with Graphpad Prism (Graphpad Software Inc.) to calculate the cell cytotoxicity 50 (CC_50_) values. The Selectivity Index (SI) relative to selected compounds was subsequently calculated as the ratio between the CC_50_ and the ED_50_ values. For MTS assays, MRC5 cells (1.5 × 10^4^ cells/well) were seeded in clear flat bottom 96-well tissue culture plates with low evaporation lids (#353072, Corning) in duplicate. After 24 h, cells were treated with different concentrations of GCV or SMs, or solvent only. A number of wells containing only DMEM and no cells were also included for background correction. Seven days post-treatment, cells were processed for measurement of cell metabolic activity using CellTiter 96^®^ AQueous One Solution Cell Proliferation Assay (G3582, Promega, Madison, WI, USA) following the manufacturer’s recommendations. After background subtraction, data were normalized to solvent-treated controls and analyzed with Graphpad Prism (Graphpad Software Inc.) For measurement of intracellular ATP levels by means of CellTiter-Glo^®^ assays (G7570, Promega), MRC5 cells were seeded in 96-Well Treated Multiwell Tissue Culture Plates, Opaque White plates (#353296, Corning). After 24 h, cells were treated with different concentrations of GCV or SMs, or solvent only. A number of wells containing only DMEM and no cells were also included for background correction. At the desired time point post-treatment, cells were processed for measurement of ATP levels following the manufacturer’s recommendations using a using a reader compatible with luminometric measurements (VICTOR X2 Multilabel Plate Reader, Perkin Elmer). After background subtraction, data were normalized to solvent-treated controls and analyzed with Graphpad Prism (Graphpad Software Inc.). The number of cells/well seeded was 1.5 × 10^4^, 4.5 × 10^3^, and 1.5 × 10^3^ cells for 24, 72, and 144 h SMs treatments, respectively.

### 2.12. Analysis of HCMV Gene Expression by Western Blotting

MRC5 cells were seeded on 6-well flat bottom plates (6 × 10^5^/well) with low evaporation lid (#353046, Corning). The following day, cells were either mock infected or infected with HCMV (strain AD169) at MOI of 2 IU/cell in DMEM at 37 °C. One hour p.i., cells were washed twice with PBS and medium containing either solvent only (0.5% DMSO), GCV, or B3 at a concentration of 6 times the ED_50_ as calculated for AD169-GFP in FRAs (16 or 50 μM, respectively), was added to each well. At different times p.i., cells were washed twice with PBS and lysed on ice with 250 μL of RIPA buffer containing protease inhibitors (Tris-HCl 50 mM, pH 7.4, 150 mM NaCl, 1% Triton X-100 (*v*/*v*), 1% sodium deoxycholate, 0.1% SDS, 1 mM EDTA, 17.4 μg/mL phenylmethylsulfonyl fluoride, 2 μg/mL aprotinin, and 4 μg/mL leupeptin) as described previously [22]. The protein content in each sample was quantified using the Micro BCA Protein Kit assay (#23235, ThermoFisher Scientific). Subsequently, 30 μg of cell lysates were diluted in Laemmli sample buffer (0.05 M Tris-HCl, pH 6.8, 0.05% Bromophenol blue, 0.1 M DTT, 10% Glycerol (*v*/*v*), 2% SDS) and boiled 5 min at 95 °C before being loaded and electrophoretically separated on 8.5 % polyacrylamide gels. Separated proteins were blotted on polyvinylidene fluoride (PVDF) membranes (RPN303F, GE Healthcare, Chicago, IL, USA). Membranes were saturated with PBS containing 0.2% Tween20 and 5% milk (*w*/*v*) and incubated with the appropriate primary and secondary antibodies diluted in PBS containing 0.2% Tween20 and 5% milk (*w*/*v*). Membranes were incubated with an enhanced chemiluminescence substrate (ECL Prime Western Blotting Detection Reagent, #RPN2236, GE Healthcare). The following antibodies, diluted in PBS containing 0.2% Tween20 and 5% milk (*w*/*v*), were used: α-His6 mAb (Sigma Aldrich, H-1029; 1:2500), α-IE1&2 mAb (P1251, Virusys Corporation, Randallstown, MD, USA; 1:10,000), α-UL44 mAb (P1202-1, Virusys Corporation; 1:100); α-pp65 mAb (CA003-1, Virusys Corporation; 1:2000); α-pp28 mAb (ab6502, Abcam, Cambridge, UK; 1:10,000), rabbit α-GADPH pAb (sc-25778, Santa Cruz Biotech, Dallas, TX, USA; 1:5000); mouse α-β-Actin mAb (A5316, Merck Millipore, Burlington, MA, USA; 1:5000); goat α-mouse (Santa Cruz Biotech, sc-2055; 1:10,000) and α-rabbit (A6154, Merck Millipore; 1:10,000) immunoglobulin Abs conjugated to horseradish peroxidase. Signals were acquired using an imaging system (Alliance Mini, Uvitech, Cambridge, UK) and quantified using Image J (NIH).

### 2.13. Analysis of HCMV DNA Replication

MRC5 cells were seeded on 24-well flat bottom plates (4 × 10^4^/well) with low evaporation lid (#353047, Corning). The following day, cells were either mock infected or infected with HCMV (TB4-UL83-EYFP) at MOI of 0.01 IU/cell in DMEM at 37 °C. Two hour p.i., cells were washed twice with PBS and medium containing either solvent only (0.5% DMSO), GCV, or B3 at a concentration of 6 times the ED_50_ as calculated for AD169-GFP in FRAs (16 or 50 μM, respectively), was added to each well. At 24, 72, and 144 h p.i., cells were detached by incubation for trypsin for 5 min. Following tryspin inactivation by addition of DMEM 10% FBS (*v*/*v*), cells were centrifugated for 5 min at 1400 rpm and stored at −80 °C until further used. Total DNA was extracted using the GenElute™ Mammalian Genomic DNA Miniprep Kits (G1N70, Merck Millipore). The levels of viral DNA were then determined by quantitative real-time PCR (qPCR) using an ABI Prism 7000 Sequence Detection System (Thermofisher Scientific) and Power SYBR Green PCR Master Mix (#4309155, Thermofisher Scientific) and were normalized to the cellular β-actin gene copies. Primers used were B2.7 F (5′-TGTTCTTCTTGGTTCATTTCC-3′) and B2.7 R (5′-CGTGTCCGGTCCTGATTC-3′), or BAF (5′-CGGGACCTGACTGACTACCTC-3′) and BAR (5′-CCATCTCTTGCTCGAAGTCCAG-3′) for detection of HCMV genomic region corresponding to the non-coding β2.7 RNA or of the human β-actin gene, respectively [53]. Raw data were used to calculate the HCMV genomes fold change, relative to each experimental condition using the ΔΔct method [54]. Values relative to each condition were further normalized to that obtained for DMSO-treated cells at 24 h p.i., and analyzed with Graphpad Prism (Graphpad Software Inc.) to calculate the *p*-value relative to each condition by means of two-way ANOVA, followed by ad hoc post test analysis for multiple comparison and Tukey correction.

## 3. Results

### 3.1. An In Silico Screening Identifies Small Molecules Potentially Interfering with ppUL44 Dimerization

We aimed at identifying small molecules (SMs) able to interfere with ppUL44 self-interaction. To this end, the published crystal structure of ppUL44 (Figure 1A) was analyzed with SiteMap to identify top-ranked potential binding pockets on the protein monomer. Our analysis identified three sites as potential binding pockets, one of which is located at the interface between the two monomers (Figure 1B). Such pocket has both a SiteScore and a druggability score (Dscore) of ≅ 0.8 according to SiteMap (0.763 and 0.797, respectively) and it is therefore potentially druggable, since a SiteScore of 0.8 has been found to accurately distinguish between drug-binding and non-drug-binding sites [44]. The key residues seem to be L87 and especially L86, which are located in a cavity formed by the residues F121, M123, M116, L93, C117, K101, T100, A118, L99, S96, D98, and P119 of the other monomer. Therefore, the center of the grid built for the docking calculation was set on M116 and such structure was used to screen about 1.3 million compounds from the ZINC database using the Glide software (Figure 1C). Eighteen of these molecules were purchased and used for further assays (Appendix A).

### 3.2. Identification of Compounds Interfering with HCMV Replication

SMs were subsequently screened for their ability to interfere with HCMV replication. In order to identify compounds active in the micromolar range, each SMs was tested at two different concentrations (10 and 100 μM), and GCV (50 μM) was included as a positive control for inhibition of viral replication. After infection and compound treatment, cells were daily monitored by microscopy for 7 days p.i. to evaluate cytopathic effect (CPE), presence of precipitates, and viral replication. At 7 days p.i., cells were lysed and the plates processed for fluorimetric quantification of the levels of viral replication relative to each condition. In parallel, mitochondrial functionality and intracellular ATP levels were quantified on uninfected cells by MTS and CellTiter Glo^®^ assays, respectively. At the lowest concentration tested (10 μM), only one compound (A4) was not soluble and caused evident cytotoxicity, whereas two compounds (B3 and B6) inhibited viral replication (Figure 2A,C). On the other hand, at the highest concentration tested (100 μM) several of the 18 SMs formed visible precipitates and caused cell death, indicating poor solubility and toxicity, while two additional compounds (B1 and C6) impaired viral replication without affecting cell viability to more than 40% (Figure 2B,D). In summary, our data showed that four SMs reduced viral replication in the absence of precipitates and evident cell cytotoxicity, to similar levels as those observed upon GCV treatment. Such inhibition was confirmed by microscopic analysis of infected cells, with a noticeable decrease in CPE and number and intensity of YFP-positive cells, to similar levels as GCV-treated cells (Appendix A and not shown).

### 3.3. Dose-Dependent Inhibition of HCMV Replication

Afterwards we determined Effective Dose (ED_50_) and Cytotoxic Concentration (CC_50_) values of each of the four SMs identified in our small-scale FRA-based screening (i.e., B1, B3, B6 and C6). To this end, we performed dose-response FRAs and MTT assays in MRC5 cells treated with increasing concentrations of each SM. Importantly, all SMs tested reproducibly inhibited viral replication in a dose-dependent manner, with ED_50_ values in the low micromolar range (Figure 3). As expected, GCV efficiently inhibited HCMV replication with a ED_50_ of 2.3 ± 0.7 μM and did not cause detectable cytotoxic effects at any concentration tested. Among the SMs tested B6 was the most potent compound and exhibited an ED_50_ slightly lower that that calculated for GCV (2.1 ± 0.6 μM). However, it was also endowed with considerable cytotoxicity (CC_50_ of ~10 μM), resulting in a poor selectivity index (SI) < 5. On the other hand, B3, the second most potent SM (ED_50_ of 4.2 ± 2.4 μM), did not cause evident cytotoxicity up to 100 μM, and therefore the SI resulted > 20. C6 also exhibited low cytotoxicity, but was significantly less efficient in inhibiting HCMV replication than the other two SMs (ED_50_ of 17.9 ± 9.6 μM), and cell morphology appeared significantly altered upon microscopic evaluation (not shown). Finally, the effect of B1 on HCMV life cycle was evident only at high concentrations (ED_50_ of 87.7 ± 14.3 μM).

To further investigate the effects of each SM on cell viability and growth, we quantified the ATP levels in MRC5 cells treated with increasing concentrations, and cultured for different times (Figure 4). Results confirmed high toxicity of compound B6, which reduced ATP intracellular levels by more than 50% at 100 μM already at 24 h post treatment and by almost 100% at 72 h post treatment. Importantly, cell treatment with C6 at 100 μM similarly reduced ATP content by more than 30% already after 24 h, whereas compounds B1 and B3 showed minimal effects at all time points analyzed. Overall, our data indicate that SM B3 might be specifically interfering with HCMV life cycle. The effect of B3 on HCMV AD169 was also assessed by PRA assays in HFF. In such experimental setting, SM B3 inhibited HCMV replication with an ED_50_ of 7.9 ± 3.5 μM (Figure 5).

### 3.4. SM B3 Inhibits Replication of a HCMV GCV-Resistant Strain

An important characteristic of an antiviral interfering with ppUL44 dimerization would be its ability to inhibit replication of viral strains resistant to the currently used antivirals. To verify if compound B3 is endowed with such ability, we compared its ED_50_ against a recombinant reporter virus AD169-GFP, and its GCV-resistant counterpart AD169-GFP26, bearing the UL97 M460I substitution [47]. To this end, MRC5 cells were infected with of either AD169-GFP or AD169-GFP26 virus at MOI of 0.05 infectious unit (IU/cell, treated with B3 or GCV for 1 week, and then viral replication was assessed by FRA. In parallel, infectious virus titers in cell culture supernatants were simultaneously quantified by VYRAs. FRA assays (Figure 6A,C and Table 1) revealed that GCV inhibited more potently replication of AD169-GFP virus than of AD169-GFP26 (ED_50_ of 2.3 ± 1.9 μM versus 21.7 ± 9.6 μM, respectively; *n* = 6), while AD169-GFP and AD169-GFP26 appeared equally sensitive to B3 (ED_50_ of 7.9 ± 2.7 μM and 17.6 ± 12.8 μM, respectively; *n* = 6). Thus, B3 appears to efficiently impair replication of both GCV-sensitive and GCV-resistant HCMV. A similar trend was observed after quantification of viral progeny in VYRA (Figure 6B,D). However, despite GCV inhibited more efficiently viral production of AD169-GFP (ED_50_ of 0.7 ± 0.4 μM; *n* = 4) as compared to AD169-GFP26 (ED_50_ of 4.7 ± 4.8 μM; *n* = 4), such difference was not statistically significant.

### 3.5. SM B3 Selectively Impairs HCMV Early and Late Gene Expression

In order to characterize the mode of action of B3, MRC5 cells were infected with AD169 at an MOI of 2 IU/cell for 2 h, treated with either vehicle alone (DMSO 0.5%), GCV (16 μM) of B3 (50 μM) and expression of immediate-early (IE1), early-late (UL44 and pp65), and late (pp28) proteins was assessed at different time points p.i. In the absence of HCMV antivirals, HCMV gene expression followed the typical pattern, with IE1 being readily detectable starting from 6 h p.i., ppUL44 from 12 h. p.i. and ppUL28 from 48 h p.i. (Figure 7 and Appendix A). As expected, neither GCV nor B3 treatment affected IE1 expression at 6 and 12 h p.i., indicating that the observed decrease in viral replication did not depend on an inhibition of activity on the major IE promoter (Figure 7A,B). Similarly, ppUL44 and pp65 levels were not affected up to 24 h p.i. (Figure 7C), confirming that IE function was not compromised. At 48 h p.i., a decrease in the expression of ppUL44 and—to a greater extent—of pp65 could be observed (Figure 7D). Importantly, both GCV and B3 treatment strongly inhibited pp28 expression at both 72 h and 96 h p.i. (Figure 7E,F). Densitometric analysis confirmed that GCV (Figure 7G) and B3 (Figure 7H) inhibited HCMV gene expression with very similar kinetics, compatible with inhibition of viral DNA synthesis and interference with an early HCMV function, possibly disrupting ppUL44 homodimers.

### 3.6. SM B3 Impairs HCMV Genome Replication

In order to investigate the effect of B3 on HCMV genome replication, MRC5 cells were infected with TB4-UL83-EYFP at an MOI of 0.01 IU/cell for 2 h, treated with either vehicle alone (DMSO 0.5%), GCV (16 μM) of B3 (50 μM) and viral genomes were quantified by qPCR at 24, 72 and 120 h p.i. In the absence of HCMV antivirals, we could detect a sharp increase of HCMV genome rapidly at 72 h and 120 h p.i. as compared 24 h p.i., indicative of active viral genome replication (Figure 8, circles). Importantly, treatment with GCV significantly reduced viral genome copy number, starting from 72 h p.i. (Figure 8, squares), consistent with inhibition of viral genome replication. Extremely similar results were obtained after treatment with B3, wherein a ~4-fold and 6-fold decrease in viral genome copy number was quantified at 72 and 120 h p.i., respectively (Figure 8, triangles). Therefore, B3 impairs HCMV life cycle by interfering with viral DNA replication.

Analysis of the predicted binding mode of B3 to ppUL44 revealed that B3 can interact with the dimerization pocket of a ppUL44 monomer in place of residues L86 and L87 of the other monomer. Indeed, B3 can establish hydrophobic interactions with ppUL44 M116, C117, A118, P119, F121, M123, L99, L93, as well as two H-bond interactions with S96, occupying a very similar position to L86 and L87 of the other monomer within the pocket itself (Figure 9). Overall, all our results are consistent with the possibility of B3 interfering with HCMV genome replication by disrupting ppUL44 homodimers.

## 4. Discussion

Disruption of PPI interaction between viral proteins is becoming an increasingly attractive strategy for the antiviral drugs development. In this context, several studies identified peptides and SMs disrupting the interaction between herpesviral DNA polymerase holoenzymes and their respective processivity factors [16,17,32,55,56]. This is the first study exploring the possibility to directly target the dimerization of HCMV DNA polymerase accessory subunit ppUL44 as an antiviral strategy. The latter represents an interesting druggable target considering the interaction interface shown in the crystal structure of ppUL44 (1-290), and the fact that single amino acid substitutions affecting dimerization in vitro also impaired dsDNA binding [23] and prevented oriLyt dependent DNA replication in trans-complementation assays [27]. Keeping this in mind, we performed a virtual screening aimed at identifying SMs inhibiting ppUL44 dimerization (Figure 1). Among the 18 SMs tested, one compound, i.e., B3, was capable of inhibiting the replication of different HCMV strains at concentrations not affecting cell growth and viability (Figure 2, Figure 3, Figure 4 and Figure 5). ED_50_ values ranged from 4.2 (FRA with TB4-UL83-EYFP) to 7.9 μM (FRA with AD169-GFP), while those determined for GCV ranged between 0.7 (VYRA) and 2.3 (FRA with AD169-GFP) μM (see Table 2). The ED_50_ values calculated here for GCV are compatible with those reported previously in the literature, with some variance being attributable to intrinsic differences between the different assays and viruses tested. For example, FRAs with the TB4-UL83-EYFP virus rely on the measurement of pp65 expression, which is expressed with an early-late kinetic, whereas the expression of the reporter gene in the AD169-GFP virus is under control of the IE promoter [47,48].

Importantly, B3 also retained antiviral activity against a GCV-resistant strain, suggesting that its mechanism of action against HCMV differs from viral DNA polymerase inhibitors (Figure 6). Although we did not formally prove here that B3 acts by disrupting the ppUL44 homodimer during viral infection, the fact that it inhibits HCMV genome replication as well as early and late gene expression starting from 48 p.i., without affecting production of immediate early viral antigens at earlier time points, in a very similar fashion to GCV (Figure 7 and Figure 8 and Appendix A) is compatible with inhibition of ppUL44 homodimerization [57,58,59].

Importantly, the effect of B3 reported here on HCMV replication is specific, since we recently demonstrated it did not affect either protein expression nor hepatitis C virus replication (HCV) in Huh7-Lunet cells [60]. Despite the fact that B3 was less potent and more toxic than GCV, it might be useful as a starting platform for hit-to-lead optimization to develop more effective compounds as it has been performed with other PPI inhibitors endowed with antiviral activity against influenza virus [61]. Future work in our laboratory will be focused at characterizing more in detail the mechanism of action of B3 against HCMV, the isolation of B3-resistant viral strains and the development of more analogs. Therefore, our results raise hopes in terms of potential use of ppUL44 dimerization inhibitors for the treatment of patients infected with drug-resistant HCMVs.

## Figures and Tables

**Figure 1 viruses-13-00941-f001:**
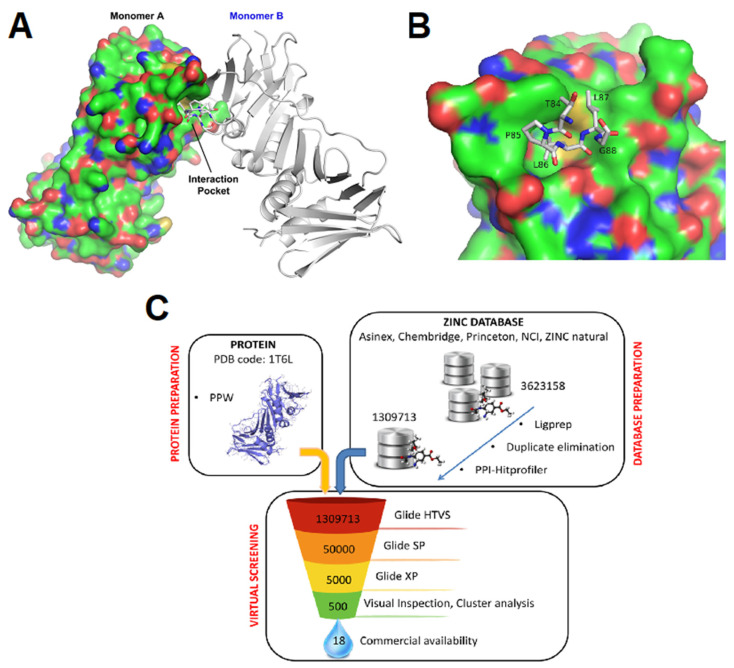
An *in silico* screening to identify SMs inhibiting ppUL44 dimerization. (**A**) Graphic representation of UL44(1-290) homodimers. One monomer is represented as surface (Monomer A), the other one as ribbons, with residues involved in dimerization being shown as sticks (Monomer B). (**B**) Inset of the homodimerization interface, with one monomer shown in surface and residues of the other monomer involved in the dimerization being shown as sticks. (**C**) Schematic overview of the virtual screening to identify SMs potentially disrupting UL44 homodimerization. The Glide software was used to dock molecules to the interface of the two monomers (PDB code: 1T6L). Three rounds of screening were performed using the High throughput virtual screening (HTVS), Standard Precision (SP), and Extra Precision (XP) docking settings. After each docking round, the top-ranked molecules in term of docking score were selected for the following round. The resulting 500 molecules were further filtered by visual inspection, cluster analysis and on the basis of their commercial availability, and 18 compounds were selected for further studies.

**Figure 2 viruses-13-00941-f002:**
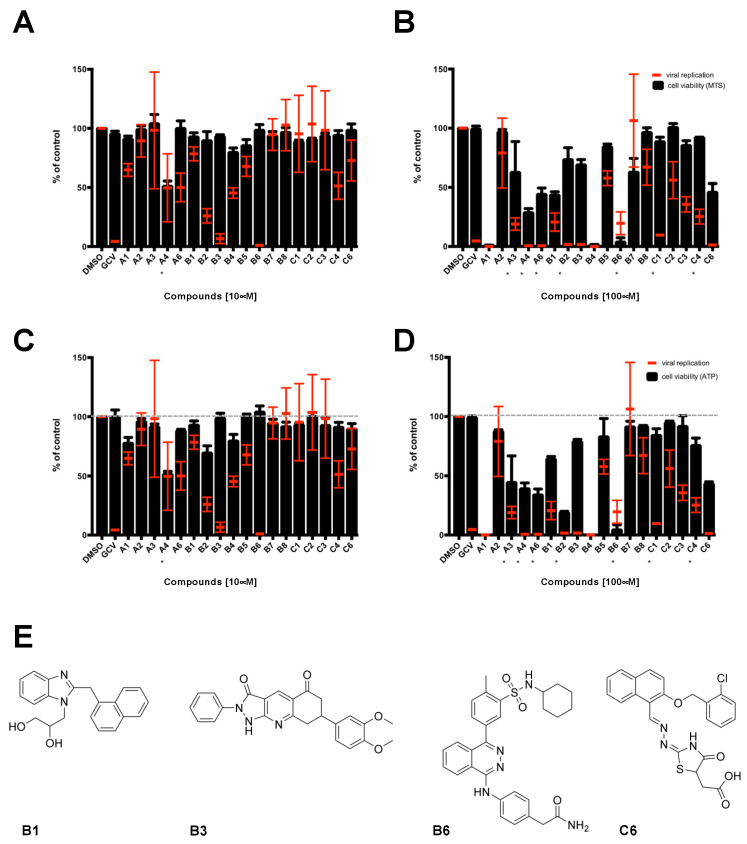
Identification of compounds interfering with HCMV replication. MRC5 cells were infected with TB4-UL83-EYFP at an MOI of 0.03 IU/cell and treated with each SM either at concentration of 10 μM (**A**,**C**) or 100 μM (**B**,**D**). In parallel, uninfected MRC5 cells were also treated for assessment of SMs cytotoxicity. Seven days post treatment, cells were processed for data acquisition and analysis as described in the Materials and Methods section. Mean YFP values relative to infected cells treated with the indicated SMs are expressed as a percentage of DMSO-treated cells (red bars). Cell viability was assessed by MTS (**A**,**B**) or Cell Titer Glo^®^ assays (**C**,**D**), and data expressed as a percentage of DMSO-treated cells (black columns). The mean + standard error of the mean (SEM) relative to 3 independent experiments is shown. * indicates the presence of precipitates. (**E**) The chemical structure of active molecules is shown.

**Figure 3 viruses-13-00941-f003:**
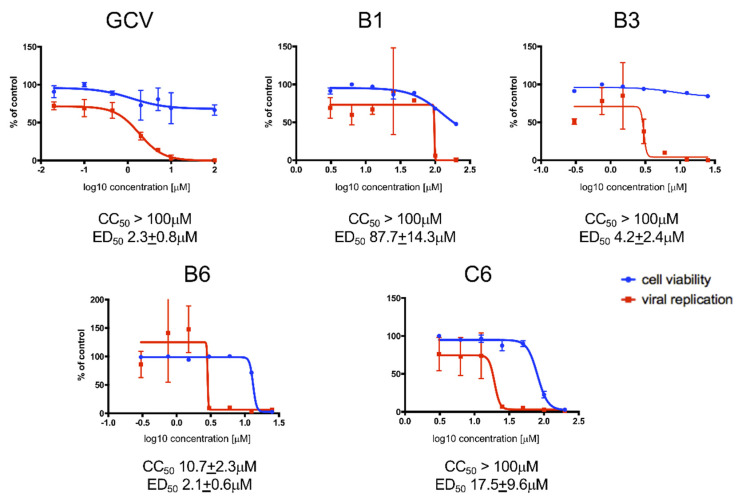
Determination of ED_50_ and CC_50_ values of SMs by FRA and MTT assay. MRC5 cells were infected with TB4-UL83-EYFP at an MOI of 0.03 IU/cell and treated with increasing concentrations of the indicated compounds. In parallel, uninfected MRC5 cells were also treated for assessment of SMs cytotoxicity. Seven days post treatment, cells were processed for data acquisition and analysis as described in the Materials and Methods section. Mean values YFP values relative to infected cells treated with the indicated compounds are expressed as a percentage of DMSO-treated cells (red squares). Cell viability was assessed by MTT assays, and data expressed as a percentage of DMSO treated cells (blue circles). For each compound, representative plots are shown, along with the cytotoxic concentration 50 (CC_50_) and effective dose 50 (ED_50_) mean values + standard deviation of the mean relative to at least 4 independent experiments.

**Figure 4 viruses-13-00941-f004:**
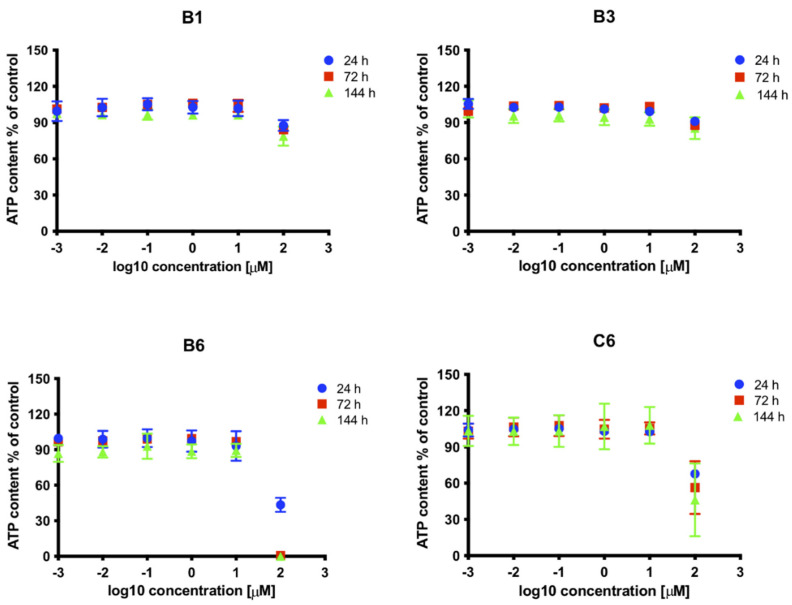
Effect on cell viability and proliferation of SMs inhibiting HCMV replication. MRC5 cells were treated with increasing concentrations of indicated compounds or with solvent only, as described in the Materials and Methods section. At the indicated time, post treatments cells were processed for intracellular ATP quantification. Data shown are the mean + standard error of the mean relative to 3 independent experiments.

**Figure 5 viruses-13-00941-f005:**
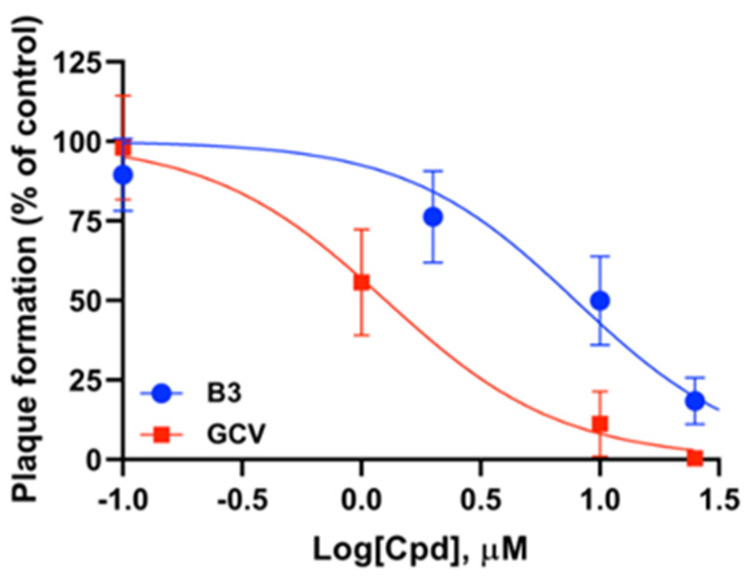
Inhibition of AD169 replication by B3 in PRA. Dose-response curves for B3 (**blue**) or GCV (**red**) were obtained by infecting HFF cells with HCMV AD169 and then treating them with different concentrations of the indicated compounds. Data shown are the means ± standard deviation of the mean relative to four independent experiments performed in duplicate.

**Figure 6 viruses-13-00941-f006:**
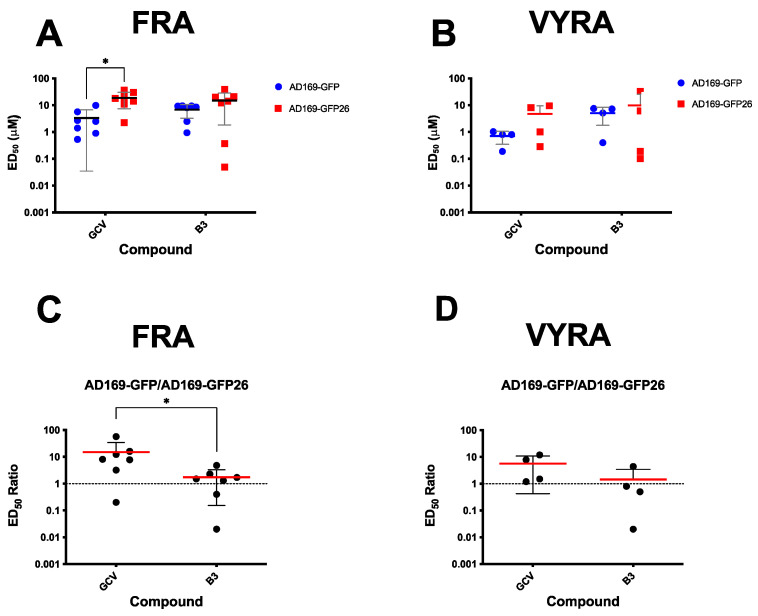
B3 efficiently inhibits replication of the GCV-resistant AD169-GFP26 virus. MRC5 were infected with either AD169-GFP virus or its GCV-resistant counterpart AD169-GFP26 at a MOI of 0.05 IU/cells and treated with increasing concentrations of the indicated compounds. At 7 days p.i., cells were lysed and plates processed for FRAs (**A**,**C**), while supernatants were collected and used for VYRAs (**B**,**D**). For the latter experiments, MRC5 cells were infected with serial dilutions of supernatants derived from infected cells. At 7 days p.i., viral titers were calculated using the TCID_50_ method. Data from both assays were used to calculate the ED_50_ relative to the two viruses (**A**,**B**) as well as the ratio between the ED_50_ calculated for AD169-GFP26 and AD169-GFP (**C**,**D**), for both GCV and B3. Data shown are single measurements, means, and standard deviation of the mean relative to at least three independent experiments (see Table 1), along with the *p*-value relative to the Student’s *t*-test the indicated groups; *: *p* ≤ 0.05.

**Figure 7 viruses-13-00941-f007:**
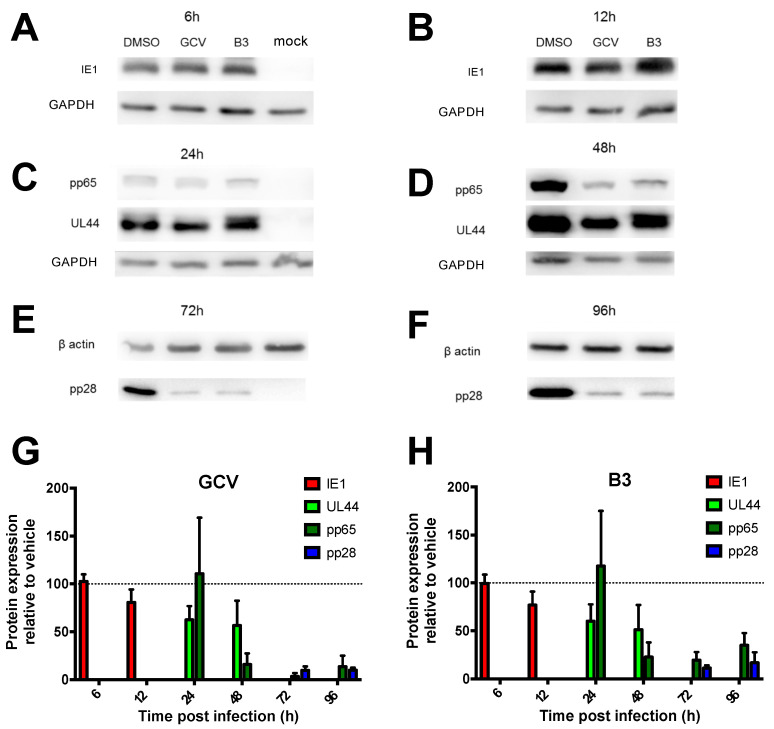
B3 specifically impairs early and late HCMV AD169 gene expression. MRC5 were infected with HCMV AD169 and treated as described in the Materials and Methods section. At the indicated time points p.i., cells were lysed and processed for Western blotting to detect the expression of the immediate early IE1 antigen (**A**,**B**; at 6 and 12 h p.i.), the early-late antigens ppUL44 and pp65 (**C**,**D**; at 24 and 48 h p.i.), and the late antigen pp28 (**E**,**F**; at 72 and 96 h p.i.). GAPDH or β-actin were also detected as loading controls. (**G**,**H**): Loading controls were used to normalize signal intensity relative to each antigen after treatment with GCV (**G**) or B3 (**H**). Lysates of mock infected cells were analyzed to verify antibody specificity. Data shown are the mean + standard deviation of the mean relative to three independent experiments.

**Figure 8 viruses-13-00941-f008:**
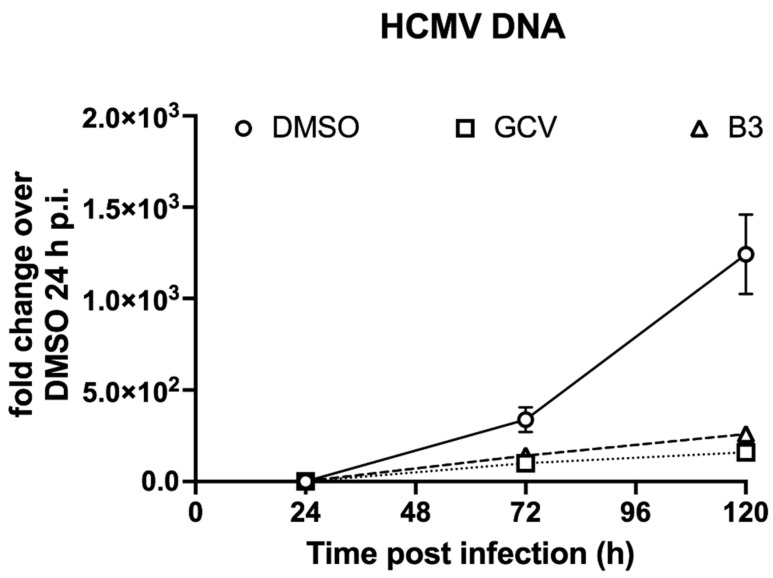
B3 impairs HCMV genome replication. MRC5 cells were infected with HCMV TB4-UL83-EYFP and treated with the indicated compounds or vehicle alone (DMSO) as described in the Materials and Methods section. At the indicated time points p.i., cells were lysed and processed for qPCR to detect the presence of viral and host cell genomic DNAs as described in the Materials and Methods section. Data are shown as HCMV DNA fold changes with respect to DMSO-treated cells at 24 h p.i. Means ± standard deviation of the mean from two independent experiments performed in quadruplicate are reported.

**Figure 9 viruses-13-00941-f009:**
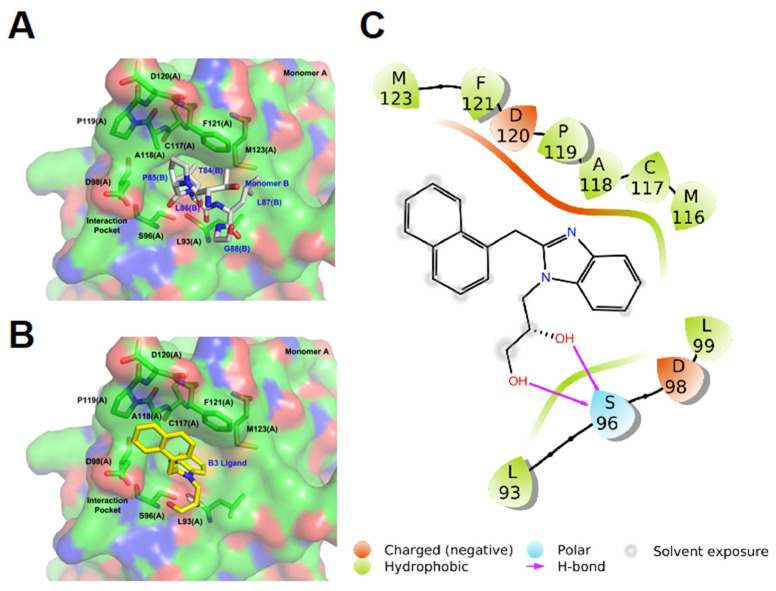
Analysis of the theoretical binding mode of B3 to ppUL44. A graphic representation of ppUL44(1-290) dimerization pocket is shown in the absence (**A**) or in the presence (**B**) of B3. One monomer is represented as surface, with residues involved in dimerization represented as green (Monomer A), or gray sticks (Monomer B), whereas B3 is shown as yellow sticks. (**C**) The 2D ligand interaction diagram of the theoretical binding mode for B3 is shown. Hydrogen bond interactions are shown as violet arrows. Positive and negative charged amino acids are represented in blue and red, respectively. Residues involved in hydrophobic or polar interactions are shown in green and light blue, respectively. Ligand-exposed fractions are indicated as a gray, circular shadow.

**Table 1 viruses-13-00941-t001:** Summary of FRA and VYA for AD169-GFP and AD169-GFP26.

	FRA (*n* = 6) ED_50_ (μM) ^1^			VYRA (*n* = 4) ED_50_ (μM)		
	AD169-GFP	AD169-GFP26	Ratio ^2^	AD169-GFP	AD169-GFP26	Ratio
GCV	2.3 ± 1.9	21.7 ± 9.6	17.4 ± 19.8	0.7 ± 0.4	4.7 ± 4.8	5.6 ± 5.2
B3	7.9 ± 2.7	17.6 ± 12.8	1.9 ± 1.6	5.1 ± 3.3	9.8 ± 15.6	1.4 ± 2.0

^1^ ED_50_, effective dose 50, the dose of compound that reduces by 50% the fluorescence (FRA) or virus titers (VYRA). ^2^ Ratio between values obtained for AD169-GFP26 and AD169-GFP. Data shown are mean and standard deviation of the mean relative to FRA (*left panel*) and VYRA (*right panels*) shown in Figure 6. The numbers between brackets indicate the number of independent experiments for each assay.

**Table 2 viruses-13-00941-t002:** Summary of ED_50_ values calculated for GCV and B3 in this study.

Assay	Virus	*n*	ED_50_ (μM)	
			GCV	B3
FRA	AD169-GFP	6	2.3 ± 1.9	7.9 ± 2.7
FRA	TB4-UL83-EYFP	6	2.3 ± 0.7	4.2 ± 2.4
PRA	AD169	4	1.3 ± 0.9	7.9 ± 3.5
VYRA	AD169-GFP	4	0.7 ± 0.4	5.0 ± 3.3

ED_50_ values from different assays (assay) and HCMV viruses (virus). The mean + standard deviation of the mean relative to the indicated number of independent experiments (*n*) is reported. FRA, fluorescence reduction assay; PRA, plaque reduction assay; VYRA, virus yield reduction assay.

## Data Availability

All data is available from the corresponding author upon request.

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
