# Peer review of "Divide et impera: An In Silico Screening Targeting HCMV ppUL44 Processivity Factor Homodimerization Identifies Small Molecules Inhibiting Viral Replication"

_viruses, 2021, doi:10.3390/v13050941_

Round 1

Reviewer 1 Report

The authors have satisfactorily addressed most of my concerns, however, some points need to be clarified before publication.

1. After the results shown in Figure 2 the authors select anti-HCMV compounds B3, B6, B1 and C6 for further characterization. However, B1 and C6 displayed near 60% cytotoxicity when used at 100 µM (Figure 2B), which is likely to be responsible for the decreased viral replication observed in cells treated with these molecules. In contrast, other compounds like C1 and C4 displayed a quite potent inhibition of viral replication and were innocuous for the cells at 100 µM but they were disqualified by the authors probably because, unlike B1 and C6, C1 and C4 formed precipitates at this high concentration. Understanding that the ideal antiviral must be non-toxic and soluble, the authors should explain better why they considered the formation of precipitates to be more decisively disqualifying than cytotoxicity. Why did they select toxic (B1 and C6) over effective and non-toxic molecules (C1 and C4) for further characterization?

2. Section 3.3. Please indicate how the selectivity index (SI) is calculated. Which is the value or interval of a good and bad SI?

3. Title of section 3.5 appears to indicate that B3 only blocks HCMV late gene expression. However, results in figure 7 show that B3 blocks the expression of the early-late genes pp65 and UL44, and accordingly, the authors state later in the discussion that B3 blocks early as well as late gene expression (line 599). Please revise the title of section 3.5.

4. Figure 8 is a nice addition, however, the description of the methods for this experiment says that qPCR data was normalized with beta-globin (Line 338) but the authors provide primer information for beta-actin. Which gene was used as reference beta-globin or beta-actin? Also, please specify which HCMV gene is amplified with primers B2.7. Furthermore, the authors say in the legend of Figure 8 that qPCR data was normalized with the value obtained for DMSO-treated infected cells at 24 h.p.i; therefore, I understand that data presented in Figure 8 is actually a fold change over the qPCR value obtained at 24 h for DMSO cells. However, the y-axis label refers to “HCMV DNA copy number”, did the author use a standard curve in their qPCR experiments to calculate the number of HCMV copies in each condition? In general, it is very confusing how these experiments were performed and analyzed. This must be clarified in the methods and the legend of Figure 8.

Author Response

The authors have satisfactorily addressed most of my concerns, however, some points need to be clarified before publication.

We thank the Reviewer for his/her positive response.

1. After the results shown in Figure 2 the authors select anti-HCMV compounds B3, B6, B1 and C6 for further characterization. However, B1 and C6 displayed near 60% cytotoxicity when used at 100 µM (Figure 2B), which is likely to be responsible for the decreased viral replication observed in cells treated with these molecules. In contrast, other compounds like C1 and C4 displayed a quite potent inhibition of viral replication and were innocuous for the cells at 100 µM but they were disqualified by the authors probably because, unlike B1 and C6, C1 and C4 formed precipitates at this high concentration. Understanding that the ideal antiviral must be non-toxic and soluble, the authors should explain better why they considered the formation of precipitates to be more decisively disqualifying than cytotoxicity. Why did they select toxic (B1 and C6) over effective and non-toxic molecules (C1 and C4) for further characterization?

We do not completely agree with the Reviewer regarding the utility of further investigate the antiviral activity of compounds active only at a concentration resulting in the formation of precipitates. Indeed compound precipitation most likely prevents accurate estimation of its concentration in the extracellular medium, and therefore in cells. Moreover, neither compound C1 nor C4 could efficiently (more than 50%) inhibit viral replication in the absence of precipitation (10 mM). On the other hand, in the case of compound B6, it is true that it exhibited significant cytotoxicity at 100 mM, but it also completely inhibit viral replication at 10 mM (Figure 2). It might have been therefore possible that such compound was endowed with an extremely potent antiviral ability (i.e. it might have been able to impair viral replication at a concentration way below the one that was tested in the original experiments of 10 and 100 mM). Obviously the selectivity index had to be experimentally calculated. Unfortunately, further experimental investigation showed that the SI for B6 to be around 10, and therefore its mechanism of action was not investigated further (Figure 3).

2. Section 3.3. Please indicate how the selectivity index (SI) is calculated. Which is the value or interval of a good and bad SI?

We thank the Reviewer for pointing this out. We accordingly modified the text in the “Materials and Methods” section 2.11, where we have inserted the following sentence: “The Selectivity Index (SI) relative to selected compounds was subsequently calculated as the ratio between the CC50 and the ED50 values“ at lines 250-252 of the revised Manuscript. Moreover, It is difficult to properly answer to the Reviewer question regarding which SI is “good”. In general, the higher is the SI of a particular compound, the more likely it is that the observed antiviral effect is specific, and not due to cytotoxicity. In the case of antivirals approved for treatment of CMV infection the SI ranges between 100 (GCV) and 15’000 (Letermovir). As we clearly point out in the Discussion section, the SI relative to B3 is not as good as those reported for approved antivirals (lines 562-8 of the revised manuscript), but it might represent a starting platform for hit-to-lead optimization.

3. Title of section 3.5 appears to indicate that B3 only blocks HCMV late gene expression. However, results in figure 7 show that B3 blocks the expression of the early-late genes pp65 and UL44, and accordingly, the authors state later in the discussion that B3 blocks early as well as late gene expression (line 599). Please revise the title of section 3.5.

We thank the Reviewer for his/her comment. Section 3.5 has now been accordingly renamed to: “SM B3 selectively impairs HCMV early-late and late gene expression.” (Line 472 of the Revised Manuscript).

4. Figure 8 is a nice addition, however, the description of the methods for this experiment says that qPCR data was normalized with beta-globin (Line 338) but the authors provide primer information for beta-actin. Which gene was used as reference beta-globin or beta-actin? Also, please specify which HCMV gene is amplified with primers B2.7. Furthermore, the authors say in the legend of Figure 8 that qPCR data was normalized with the value obtained for DMSO-treated infected cells at 24 h.p.i; therefore, I understand that data presented in Figure 8 is actually a fold change over the qPCR value obtained at 24 h for DMSO cells. However, the y-axis label refers to “HCMV DNA copy number”, did the author use a standard curve in their qPCR experiments to calculate the number of HCMV copies in each condition? In general, it is very confusing how these experiments were performed and analyzed. This must be clarified in the methods and the legend of Figure 8.

We thank the Reviewer for his/her extremely useful comment. We have now corrected the qPCR methods section to mention that beta actin was used for normalization (Line 309 of the revised Manuscript). We also specify that the HCMV-specific primers target the genomic region corresponding to the non-coding β2.7 RNA (Line 312 of the revised Manuscript). Furthermore, since data were normalized to those obtained for DMSO-treated infected cells at 24 h.p.i, this is now clearly stated in the Material and Methods section (Lines 310-312 of the revised Manuscript). Moreover, Figure 8 has been modified to include the title “HCMV DNA” , and the y-axis title as been changed to “fold change over DMSO 24 h p.i.”. Finally, the Figure Legend has now been modified to “Figure 8. B3 impairs HCMV genome replication. MRC5 cells were infected with HCMV TB4-UL83-EYFP and treated with the indicated compounds or vehicle alone (DMSO) as described in the Materials and Methods section. At the indicated time points p.i., cells were lysed and processed for qPCR to detect the presence of viral and host cell genomic DNAs as described in the Materials and Methods section. Data are shown as HCMV DNA fold changes with respect to DMSO-treated cells at 24 h p.i. Means + standard deviation of the mean from two independent experiments performed in quadruplicate are reported.”

Reviewer 2 Report

The authors have done an excellent job of revision, particularly with the addition of new data on the effect of the B3 compound on viral DNA copy during infection, and clarification of the residues involved in ppUL44 homodimer interactions.

Some minor fixes:

Line 19: rephrase

Lines 26 and 92: “consistent”, not “consistently”

Line 47: “of”, not “on”

Line 511: “was capable of”, not “was able of”

Line 513-14: “ranged”, not “were comprised”

Line 517: “relies” not “rely”

Line 541: “Despite the fact that B3 was less potent…”

Author Response

The authors have done an excellent job of revision, particularly with the addition of new data on the effect of the B3 compound on viral DNA copy during infection, and clarification of the residues involved in ppUL44 homodimer interactions.

We thank the Reviewer for his/her very positive feedback.

Some minor fixes:

Line 19: rephrase

We thank the reviewer for helping us improving the clarity of our manuscript. We  have modified the text that now reads “Since dimerization of HCMV DNA polymerase processivity factor ppUL44 plays an essential role in the viral life cycle, being required for oriLyt-dependent DNA replication, it can be considered a potential therapeutic target. We therefore performed an in silico screening and selected 18 small molecules (SMs) potentially interfering with ppUL44 homodimerization.” (Lines 16-18 of the revised manuscript).

Lines 26 and 92: “consistent”, not “consistently”

Line 47: “of”, not “on”

Line 511: “was capable of”, not “was able of”

Line 513-14: “ranged”, not “were comprised”

Line 541: “Despite the fact that B3 was less potent…”

We have modified the text according to the Reviewer’s suggestion.

Line 517: “relies” not “rely”

We modified “FRA rely” to “FRAs rely”

This manuscript is a resubmission of an earlier submission. The following is a list of the peer review reports and author responses from that submission.

Round 1

Reviewer 1 Report

In this manuscript, the authors use an in silico approach to identify inhibitors of the dimerization of the HCMV processivity factor ppUL44, as a potential strategy to block HCMV infection. Then, the authors test in vitro the anti-HCMV activity of selected candidate compounds. Although the mechanism of action of the most successful compound B3 is not demonstrated, the authors openly discuss this limitation of their study in the Discussion. In the absence of a licensed anti-HCMV vaccine and given the side effects and viral resistance of ganciclovir, the most widely used anti-HCMV drug, I find the results presented in this manuscript timely and relevant. In addition, the paper is well written and is easy to read. However, I have some concerns, comments and suggestions that the authors should address before the paper can be accepted for publication.

  1. Authors should frame in Figure 1A the protein region represented in Figure 1B. Also, in the description of panel A in the legend of Figure 1 it is said that “residues involved in the dimerization being shown as sticks”. However, I do not see any stick representation in panel 1A.

  1. Authors should provide in Material and Methods or Results the scale of the druggability score and explain which range is considered “potentially druggable”, as stated in line 207.

  1. After reading the introduction and the title of section 3.1 it is clear that the main target of the authors is to identify small molecules capable of interfering with ppUL44 dimerization. However, later in line 204, the authors say that they analyzed the published ppUL44 structure to identify “receptor binding sites on the protein monomer”. This is confusing. Which is the receptor in this case? Please explain.

  1. Authors should include a new methods section explaining the source and the culture conditions of all the cell lines used in their study, and another section to describe how their viral stocks (TB4-UL83-YFP, AD169,…) were prepared, grown, purified, and titrated.

  1. In line 238 the authors say “compounds (B1 and C6) impaired viral replication without markedly affecting cell viability”. However, Figure 2B shows that only about 40% of the cells survived to 100 µM of these compounds, which seems a quite significant cytotoxicity. Please rephrase.

  1. Line 242, “data not shown”, if available, CPE and YFP positivity data should be included.

  1. Line 264, please define “SI”.

  1. In line 265 the authors declare that B3 did not cause evident cytotoxicity up to 100 µM. However, the B3 panel in Figure 3 only presents cytotoxicity information up to 30 µM. Please change the x-axis scale of this graph to include data at 100 µM and convince the reader that B3 was innocuous and this concentration as stated in line 265. In fact, the scale of the x-axis is different in all graphs. Please use the same scale for all compounds. Also, the units in the x-axis labels and the CC50 and ED50 values are indicated as “αM”, which I assume is a typo. Please correct.

  1. In Figure 3, B1, B3, B6, and C6, display a cell viability close to 100% when used at 10 µM, but their cell viability at this same concentration in Figure 2A is 50% or less (except for C6). How do the authors explain these inconsistencies?

  1. Line 268, “not shown”, if available please include these data.

  1. Figure 4. Please present the x-axis of the B3 panel in the same scale and format used for the other 3 panels.

  1. Most anti-HCMV experiments in the paper are performed at the single time point of 7 days post infection. The authors should at least compare the anti-HCMV activity of a fixed dose (e.g. 10X ED50) of B3 with that of ganciclovir over time (1, 2, 3,…, 7 days post infection) to better understand and compare the kinetics of these anti-HCMV compounds.

  1. Figure 6. Although by FRA (panels A and C) it is clear that GCV is more efficient against AD169-GFP than AD169-GFP26, this is not as clear by VYRA (panels B and D). In fact, Figure 6D shows 2 points with a ED50 AD169-GFP/AD169-GFP26 ratio close to 1 (no different susceptibility to GCV), a ratio that is only brought up by a single point that could possibly be an outlier. The authors should increase the number of data points to improve the statistical rigor or tone down their conclusions from these experiments. Also, the reference ED50 ratio value is 1, therefore the y-axes of Figures 6C and 6D should be represented in log scale or any other scale that allows the reader to clearly visualize the position of this reference value of 1 on the y-axis.

  1. Figure 7 and supplementary figure S2 should include western blots panels for all viral proteins analyzed at time 0h or mock-infected cells.
  1. The figure legend of supplementary figure S2 does not describe panel A, B and C. Please explain in the legend what the different panels correspond to.

  1. GAPDH is misspelled in Figure 7. Please correct.

  1. Line 341, change “Figure 8” for Figure 7; Line 348 change “Figure 8G” and “Figure 8H” for Figure 7G and 7H, respectively. Change ppUL28 to pp28.

Author Response

In this manuscript, the authors use an in silico approach to identify inhibitors of the dimerization of the HCMV processivity factor ppUL44, as a potential strategy to block HCMV infection. Then, the authors test in vitro the anti-HCMV activity of selected candidate compounds. Although the mechanism of action of the most successful compound B3 is not demonstrated, the authors openly discuss this limitation of their study in the Discussion. In the absence of a licensed anti-HCMV vaccine and given the side effects and viral resistance of ganciclovir, the most widely used anti-HCMV drug, I find the results presented in this manuscript timely and relevant. In addition, the paper is well written and is easy to read. However, I have some concerns, comments and suggestions that the authors should address before the paper can be accepted for publication.

Authors should frame in Figure 1A the protein region represented in Figure 1B. Also, in the description of panel A in the legend of Figure 1 it is said that “residues involved in the dimerization being shown as sticks”. However, I do not see any stick representation in panel 1A.

We thank the Reviewer for the suggestion. In the revised Figure 1, the protein region shown is now clearly indicated in Figure 1A, and residues involved in the dimerization have been shown as sticks.

Authors should provide in Material and Methods or Results the scale of the druggability score and explain which range is considered “potentially druggable”, as stated in line 207.

We apologize for this possible point of confusion. We have modified the Results section to include both SiteScore and Dscore, added an appropriate reference for the 0.8 threshold. The text now reads “Such pocket has a SiteScore of 0.80 and a druggability score of @ 0.8 according to SiteMap (0.763 and 0.797, respectively) and it is therefore potentially druggable, since a SiteScore of 0.8 has been found to accurately distinguish between drug-binding and non-drug-binding sites [44].”

After reading the introduction and the title of section 3.1 it is clear that the main target of the authors is to identify small molecules capable of interfering with ppUL44 dimerization. However, later in line 204, the authors say that they analyzed the published ppUL44 structure to identify “receptor binding sites on the protein monomer”. This is confusing. Which is the receptor in this case? Please explain.

We apologize to the Reviewer for this point of possible confusion, We have accordingly modified the sentence in the revised manuscript, which now reads “binding pockets on the protein monomer”.

Authors should include a new methods section explaining the source and the culture conditions of all the cell lines used in their study, and another section to describe how their viral stocks (TB4-UL83-YFP, AD169,…) were prepared, grown, purified, and titrated.

We thank the Reviewer for pointing out this shortcoming. Relevant information has now been added to the Materials and Methods section, see "Section 2.4 Cell lines." and "Section 2.5 Viruses, viral stocks preparation and titration."

In line 238 the authors say “compounds (B1 and C6) impaired viral replication without markedly affecting cell viability”. However, Figure 2B shows that only about 40% of the cells survived to 100 µM of these compounds, which seems a quite significant cytotoxicity. Please rephrase.

To deal with this and additional comments we have performed new cytotoxicity assays using MTS rather than MTT. Such assays similarly measure mitochondrial activity but the product, unlike formazan, is completely soluble, making its solubilisation unnecessary before data acquisition. Therefore, readings result more robust and reliable. We also performed a quantification of intracellular ATP levels by means of luminometric assays, to further strengthen our data. Results are shown in the new Figure 2 and show that both treatment with B1 and C6 at 100 mM resulted in approximatively 50% reduction of cell metabolic activity. Consistently, the text has been modified accordingly “two compounds (B1 and C6) impaired viral replication without affecting cell viability by more than 40% (Figure 2B)”

Line 242, “data not shown”, if available, CPE and YFP positivity data should be included.

According to the Reviewer’s suggestion we have included micrographs clearly showing differences in YFP expression in Supplementary Figure S2. We did not acquire at that time bright field images, so that data relative to CPE are still referred as “not shown”.

Line 264, please define “SI”.

We thank the reviewer for pointing this out, the definition of selectivity index (SI) has been added to the text (line 374 of the revised manuscript)

In line 265 the authors declare that B3 did not cause evident cytotoxicity up to 100 µM. However, the B3 panel in Figure 3 only presents cytotoxicity information up to 30 µM. Please change the x-axis scale of this graph to include data at 100 µM and convince the reader that B3 was innocuous and this concentration as stated in line 265. In fact, the scale of the x-axis is different in all graphs. Please use the same scale for all compounds. Also, the units in the x-axis labels and the CC50 and ED50 values are indicated as “αM”, which I assume is a typo. Please correct.

We thank the reviewer for pointing the typo, which we promptly corrected (and was due to the dreadful "mac to PC pdf conversion"). We present cell cytotoxicity data up to 100 uM in Figure 4 (and of course Figure 2), relative to compounds B1, B3, B6 and C6. Experiments in Figure 3 were performed at concentrations sufficient to completely abolish viral replication in such assays. Although we could repeat them at higher concentrations, their utility would be dubious, considering data shown in Figure 2B, 2D and Figure 4, whereby the effect of B3 on cell viability is tested up to 100 uM, either by MTS and Celltiter Glo assays, respectively.

In Figure 3, B1, B3, B6, and C6, display a cell viability close to 100% when used at 10 µM, but their cell viability at this same concentration in Figure 2A is 50% or less (except for C6). How do the authors explain these inconsistencies?

We have replaced the MTT assays shown in Figure 2 with more robust MTS assays, wherein resulted that cell viability of cells treated with the indicated compounds at 10 uM compounds is not affected (see Figure 2B). As mentioned above, we have also included new data from experiments aimed at measuring ATP intracellular levels (see Figure 2D, Figure 4), which further strengthen our results. Concerns on effects of cytotoxicity should also be relieved by data shown in Figure 4. We thank the Reviewer for pointing out such discrepancy in our data.

Line 268, “not shown”, if available please include these data.

Unfortunately, microscopic evaluation is not documented by digital images, therefore that data cannot be shown. If the Reviewer prefers, we could remove the sentence.

Figure 4. Please present the x-axis of the B3 panel in the same scale and format used for the other 3 panels.

We have uniformed the x-axis for B3 to that used for other compounds. We thank the Reviewer for noticing this issue.

Most anti-HCMV experiments in the paper are performed at the single time point of 7 days post infection. The authors should at least compare the anti-HCMV activity of a fixed dose (e.g. 10X ED50) of B3 with that of ganciclovir over time (1, 2, 3,…, 7 days post infection) to better understand and compare the kinetics of these anti-HCMV compounds.

Indeed, most assays used in our study to investigate antiviral activity of compounds measure differences in viral spread and production, and therefore are correctly performed at 7 days post infection, in order to allow differences between treated and untreated cells to become visible. This is the case for plaque reduction assays, as well as fluorescence reduction assays using AD169-GFP and TB4-pp65-YFP. However, data shown in Figure 7, whereby we individually assess the levels of several viral antigens which are expressed with different kinetics during the viral life cycle, clearly demonstrate that the action of B3 and GCV is kinetically comparable, reducing viral antigen expression starting from 48h post infection. Clearly, no effect on IE gene expression was measured at 6 and 12 h post infection, while strong inhibition of early/late and late gene expression was measurable starting from 48h post infection. In order to satisfy the Reviewer’s (and ours) curiosity, these data are now corroborated by new qPCR experiments whereby the effect of B3 on HCMV DNA replication is clearly shown (see new Figure 8), starting from 72 h post infection, with -once again - kinetics extremely similar to GCV. We thank the Reviewer for his/her suggestion which contributed to strongly enhance the soundness of out work. 

Figure 6. Although by FRA (panels A and C) it is clear that GCV is more efficient against AD169-GFP than AD169-GFP26, this is not as clear by VYRA (panels B and D). In fact, Figure 6D shows 2 points with a ED50 AD169-GFP/AD169-GFP26 ratio close to 1 (no different susceptibility to GCV), a ratio that is only brought up by a single point that could possibly be an outlier. The authors should increase the number of data points to improve the statistical rigor or tone down their conclusions from these experiments. Also, the reference ED50 ratio value is 1, therefore the y-axes of Figures 6C and 6D should be represented in log scale or any other scale that allows the reader to clearly visualize the position of this reference value of 1 on the y-axis.

We completely agree with the Reviewer, and apologize for not having stated this clearly enough in the original submission that in FRA assays no significant difference by virus yield assays after infection with either wild-type or GCV resistant virus is observed after treatment with GCV. This is most likely due to the higher variability of the assay as compared to the much more robust FRA. For this reason, as suggested by the Reviewer, we repeated once the experiment. However, the difference  in the results still did not reach statistical significance. Since we are not convinced that it is completely correct to repeat experiments in order to just reach statistical significance, we did not further repeat the experiment. The lack of significance of the VYRA is now clearly mentioned in the text and, as requested by the Reviewer, a log10 scale is used in the y axes of panels C and D, along with a horizontal line in correspondence to a ED50 AD169-GFP/AD169-GFP26 ratio of 1.

Figure 7 and supplementary figure S2 should include western blots panels for all viral proteins analyzed at time 0h or mock-infected cells. 

We thank the Reviewer for his/her suggestion, and have modified Figure 7 as suggested in order to include western blot images of mock infected cells for all tested antigens.

The figure legend of supplementary figure S2 does not describe panel A, B and C. Please explain in the legend what the different panels correspond to.

We thank the reviewer for noticing such issue, Figure legend has now been corrected accordingly.

GAPDH is misspelled in Figure 7. Please correct.

We thank the Reviewer for noticing such issue, which has now been corrected.

 Line 341, change “Figure 8” for Figure 7; Line 348 change “Figure 8G” and “Figure 8H” for Figure 7G and 7H, respectively. Change ppUL28 to pp28.

We thank the Reviewer for noticing such issues, which have now been corrected.

Reviewer 2 Report

Line 53: please correct "divided in" to "divided into"

Line 338: What is an "MOI of 2 IU/cell"? Did you mean pfu/cell?

Author Response

Line 53: please correct "divided in" to "divided into"

We thank the Reviewer for pointing this out. The sentence has been corrected accordingly.

Line 338: What is an "MOI of 2 IU/cell"? Did you mean pfu/cell?

We apologize for not having explained in the original submission the exact procedure which we used for virus stocks titration, which was made by counting IE1/2 positive foci by immunofluorescence rather than by plaque assays. Therefore, we always express the used MOIs as infectious units (IU)/cell. The materials and methods section has been modified accordingly to explain how viral stocks were obtained and titred.

Reviewer 3 Report

This paper follows up on a series of papers from this group that have investigated the hypothesis that homodimerization of ppUL44 is required for replication of HCMV DNA in infected cells, and is therefore a potentially druggable therapeutic target.  Previous studies identified specific residues in ppUL44 that mediate homodimerization, interaction with the viral DNA polymerase UL54, and OriLyt-dependent DNA replication.  Here the authors have used an in silico drug screening strategy to identify a small molecule inhibitor that purportedly disrupts ppUL44 dimerization and HCMV replication.  They use ganciclovir as a positive control for inhibition of replication, and they analyze the ability of this compound to inhibit replication of a GCV-resistant strain of HCMV.

Strengths:

The paper identifies a compound that appears to modestly inhibit HCMV virus production and expression of late gene proteins, albeit less effectively than GCV. 

Weaknesses:

  1. The authors state that the B3 compound inhibits HCMV replication on the basis of a fluorescence reduction assay (FRA). This assay measures fluorescence due to expression of a UL83-YFP fusion protein encoded by the virus. There is no reference describing this virus.  Although they state that they are measuring viral replication, they are actually measuring expression of pUL83, which is not an essential protein.  This should be made explicit in the text. 
  2. The authors suggest that the compound works by inhibiting homodimerization of ppUL44 and DNA replication. They do not present direct evidence to support this.  The authors acknowledge that they did not analyze dimerization.  However, the paper would be significantly strengthened by using a quantitative assay for the effect of the compound on viral DNA copy number.
  3. In Fig. 8 the authors present a model that predicts the residues in ppUL44 that may be targeted by the B3 compound. These appear to be different from the residues that were previously identified as mediating the functions of ppUL44.  The authors do not comment on this difference.
  4. A comparison of the effect of the B3 compound versus mutation of the residues shown to be important in mediating ppUL44 dimerization (L86A/L87A) would have been informative.
  5. The results of the two different assays in Fig. 6 comparing the effects of GCV and B3 on replication of GCV-resistant virus appear to be inconsistent. The authors do not comment on this discrepancy.

Minor points:

  1. Line 37 states that packaging of the virus occurs in the nucleus. Encapsidation of the DNA occurs in the nucleus, but assembly of viral particles occurs in the cytoplasm.
  2. Typo in Line 376.

Author Response

This paper follows up on a series of papers from this group that have investigated the hypothesis that homodimerization of ppUL44 is required for replication of HCMV DNA in infected cells, and is therefore a potentially druggable therapeutic target.  Previous studies identified specific residues in ppUL44 that mediate homodimerization, interaction with the viral DNA polymerase UL54, and OriLyt-dependent DNA replication.  Here the authors have used an in silico drug screening strategy to identify a small molecule inhibitor that purportedly disrupts ppUL44 dimerization and HCMV replication.  They use ganciclovir as a positive control for inhibition of replication, and they analyze the ability of this compound to inhibit replication of a GCV-resistant strain of HCMV.

Strengths:

The paper identifies a compound that appears to modestly inhibit HCMV virus production and expression of late gene proteins, albeit less effectively than GCV. 

Weaknesses:

The authors state that the B3 compound inhibits HCMV replication on the basis of a fluorescence reduction assay (FRA). This assay measures fluorescence due to expression of a UL83-YFP fusion protein encoded by the virus. There is no reference describing this virus.  Although they state that they are measuring viral replication, they are actually measuring expression of pUL83, which is not an essential protein.  This should be made explicit in the text. 

We apologize for not having described the virus in more detail. The virus is now fully described and the paper originally describing it is explicitly cited. Furthermore, we would like to highlight that data in our original submission did not solely rely on pp65 expression, but also on plaque reduction assays and Western blotting using the widely used laboratory strain AD169 (see Figure 5 for plaque reduction assays and Figure 8 for Western blot assays) as well as on FRAs and VYRAs with recombinant virus AD169-GFP, therefore implying effect on viral particles production and hence viral replication, rather than only on expression on pp65. We also added new data quantifying the effect of B3 on HCMV genome replication, clearly showing a marked decrease as compared to vehicle treated cells (see new Figure 8). We thank the reviewer for pointing out such issue and helping us to significantly improve the strength of our data.

The authors suggest that the compound works by inhibiting homodimerization of ppUL44 and DNA replication. They do not present direct evidence to support this.  The authors acknowledge that they did not analyze dimerization.  However, the paper would be significantly strengthened by using a quantitative assay for the effect of the compound on viral DNA copy number.

According to the Reviewer’s suggestion, new experimental data have been generated to include analysis of the effects of B3 on HCMV genome replication. New data are shown in Figure 8, and show very similar kinetics with the control drug GCV. We sincerely thank the reviewer for his/her very helpful suggestion which allowed to increase the soundness of our work.

In Fig. 8 the authors present a model that predicts the residues in ppUL44 that may be targeted by the B3 compound. These appear to be different from the residues that were previously identified as mediating the functions of ppUL44.  The authors do not comment on this difference.

We apologize with the reviewer for the poor clarity of our text and figures when dealing with this very important point. Indeed, B3 interacts with ppUL44 binding pocket one monomer in a very similar fashion to residues L86 and L87 from the other monomer. Unfortunately, this was not explained and presented in a sufficiently clear fashion in the original submission of our manuscript. We have now modified Figure 8 (Figure 9 in the revised manuscript) and the text in the results section accordingly. In particular, we have added a new panel to original Figure 8 (now Figure 9), showing one monomer as surface and the other monomer as ribbons, whereas another panel shows only one monomer as surface and B3 instead of the other ppUL44 monomer, which should be displace by B3. Indeed, when describing the UL44 dimerization interface (Figure 1), we mentioned that “The key residues seem to be L87 and especially L86, which are located in a cavity formed by the residues F121, M123, M116, L93, C117, K101, T100, A118, L99, S96, D98 and P119 of the other monomer L86 and L87”. Now in the description of B3 binding mode (Results section), we also added the following sentence “Analysis of the predicted binding mode of B3 to ppUL44 revealed that B3 can interact with the dimerization pocket of a ppUL44 monomer, in place of residues L86 and L87 of the other monomer. Indeed, B3 can establish hydrophobic interactions with ppUL44 M116, C117, A118, P119, F121, M123, L99, L93 as well as two H-bond interactions with S96, occupying a very similar position to L86 and L87 within the pocket itself (Figure 8)”. As we hope it is clearer now, B3 inserts in the cavity which normally would accommodate L86 and L87, thus potentially explaining its antiviral properties. Former Figure 8 (Figure 9 in the revised manuscript) has been further enhanced for clarity by separating panel A, containing the crystal structure of one monomer of UL44 in complex with B3 and the other monomer, in two independent panels, whereby one UL44 monomer is shown either in complex with the other monomer (panel A) or with B3 (panel B).

A comparison of the effect of the B3 compound versus mutation of the residues shown to be important in mediating ppUL44 dimerization (L86A/L87A) would have been informative.

We agree with the Reviewer that testing B3 activity on a L86A/L87A mutant virus would be extremely interesting, however this approach is severely limited experimentally by the fact that the L86A/L87A completely abolished oriLyt dependent DNA replication. Therefore, we hypothesize that a L86A/L87A recombinant virus would not be viable.

The results of the two different assays in Fig. 6 comparing the effects of GCV and B3 on replication of GCV-resistant virus appear to be inconsistent. The authors do not comment on this discrepancy.

The Reviewer is correct. As also pointed out by Reviewer 1, there is indeed an inconstancy between results in FRA and VYRA. In particular, despite all data have the same general trend (i.e., GCV is less efficient in inhibiting wild-type vs. GCV resistant virus), results are statistically significant only in FRA. This is mainly due to the greater variability that we observed in VYRA and which is intimately linked to the assay readout. The lack of statistical significance in case of VYRA is now mentioned in the text. Clearly FRA, which relies on automatic fluorescence measurements by an instrument, is more robust compared to VYRA, which might depend on operator’s counting of productively infected cells.

Minor points:

Line 37 states that packaging of the virus occurs in the nucleus. Encapsidation of the DNA occurs in the nucleus, but assembly of viral particles occurs in the cytoplasm.

We are very grateful to the reviewer for pointing out such mistake, which has been corrected in the revised version of our manuscript. We meant that genomes are packaged into the capsid.

 Typo in Line 376.

We thank the Reviewer for pointing out such issue, which has now been corrected.